

# The high-temperature expansion of the thermal sunset

**Andreas Ekstedt[1]★ and Johan Löfgren[2]†**

**1** Institute of Particle and Nuclear Physics, Charles University, Prague, Czech Republic
**2** Department of Physics and Astronomy, Uppsala University, Uppsala, Sweden

★ andreas.ekstedt@ipnp.troja.mff.cuni.cz, † johan.lofgren@physics.uu.se

## Abstract

We give a prescription for calculating the high-temperature expansion of the thermal sunset integral to arbitrary order. We derive all terms odd in $T$, and rederive previous results up to $\mathcal{O}(T^0)$ for both bosonic and fermionic thermal sunsets in dimensional regularisation. We perform analytical and numerical cross-checks. Intermediate steps involve integrals over three Bessel functions.

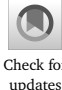

# 1 Introduction

Finite-temperature field theory has a wide range of applications, from phase transitions to the inner workings of neutron stars. Although the field was established last century, the techniques evolve constantly to deal with an ever-increasing demand for precision—and fresh applications. Finite temperature calculations are perturbative whenever possible, though lattice calculations are also viable.

Yet perturbative calculations are arduous and often involve many mass scales. Fortunately, many applications feature a hierarchy between particle masses and the temperature: $T^2 \gg X$; a high-temperature expansion is applicable. Such expansions are known to all orders at one loop, but not at higher loop orders where the computations are more intricate.

To circumvent this problem one can resort to numerical evaluations, which is a viable strategy for some cases. Yet this leaves something to be desired when the temperature follows a strict power counting. Not to mention the computational complexity of numerical integrals; especially if numerous mass scales are present.

On the analytical side, there has been much progress by using the method of Integration-By-Parts (IBP) [1–4]. With IBP relations it is possible to reduce any massless 2-loop integral into 1-loop integrals that are fully known [4]. There are also a number of results at higher loop orders [3,4]. For massive 2-loop integrals there are some partial results [5].

In this paper we focus on the high-temperature expansion of the 2-loop bosonic and fermionic thermal sunset integrals with arbitrary masses. The leading $T^2$ contribution, in both the bosonic and the fermionic case, are long known [6,7]. These massive sum-integrals are important for accurate studies of the electroweak phase transition.

We provide an organizational framework to calculate the high-temperature expansion of the sunset to arbitrary order, and calculate all terms that are non-analytic in the squared masses, starting at order $T$. The remaining analytic terms can all be given by IBP relations, though we rederive the $T^0$ terms using an alternative method. As intermediate steps, sums and integrals of three Bessel functions are given. To ensure the validity of the results we perform theoretical and numerical cross-checks.

## 2 Results

### 2.1 Bosonic sunset

The thermal bosonic sunset for three arbitrary squared masses $X$, $Y$, $Z$ is

$$\mathbf{I}(X,Y,Z) \equiv T^2 \sum_{n_p,n_q,n_l} \int_{\vec{p},\vec{q},\vec{l}} \delta(\vec{p}+\vec{q}+\vec{l})\delta_{n_p+n_q+n_l,0}$$

$$\times \frac{1}{\vec{p}^2 + X + (2\pi n_p T)^2} \frac{1}{\vec{q}^2 + Y + (2\pi n_q T)^2} \frac{1}{\vec{l}^2 + Z + (2\pi n_l T)^2}, \tag{2.1}$$

with the measure $\int_{\vec{p}} = \left(\frac{\mu^2 e^\gamma}{4\pi}\right)^\epsilon \int \frac{d^d\vec{p}}{(2\pi)^d}$, $d = 3 - 2\epsilon$; we use dimensional regularisation in the $\overline{\text{MS}}$ scheme, and take $\mu$ as the $\overline{\text{MS}}$ scale. More compactly,

$$\mathbf{I}(X,Y,Z) \equiv \sum_{P,Q} \frac{1}{P^2+X} \frac{1}{Q^2+Y} \frac{1}{(P-Q)^2+Z}, \tag{2.2}$$

$$P^2 = \vec{p}^2 + (2\pi T n_p)^2. \tag{2.3}$$

Expand the sunset as

$$\mathbf{I}(X,Y,Z) = \frac{I_{-2}(X,Y,Z)}{\epsilon^2} + \frac{I_{-1}(X,Y,Z)}{\epsilon} + I(X,Y,Z) + \mathcal{O}(\epsilon), \tag{2.4}$$

where the divergent contributions are known to all orders in $T$,

$$I_{-2}(X,Y,Z) = -\frac{1}{(16\pi^2)^2} \frac{X+Y+Z}{2}, \tag{2.5}$$

$$I_{-1}(X,Y,Z) = \frac{1}{16\pi^2}\left(A(X)+A(Y)+A(Z) - \frac{1}{16\pi^2}\frac{X+Y+Z}{2}\right). \tag{2.6}$$

Here $A(X)$ is the finite piece of the bosonic 1-loop bubble integral, see equation (3.10).

The finite piece $I(X,Y,Z)$ to order $T^2$ is long known [6–8], and the $T^0$ piece can be inferred from [4]. We derive all terms with odd powers of $T$ in section 4.1, and the $T^0$ terms in section 4.2. In summary,

$$
\begin{aligned}
I(X,Y,Z) = {} & \frac{T^2}{16\pi^2}\left(\log\left[\frac{\mu}{\sqrt{X}+\sqrt{Y}+\sqrt{Z}}\right] + \frac{1}{2}\right) \\
& - \frac{T}{64\pi^3}\left(\left(\sqrt{X}+\sqrt{Y}+\sqrt{Z}\right)\left(\log\left[\frac{e^{2\gamma}\mu^2}{16\pi^2 T^2}\right]+2\right)\right. \\
& \qquad\quad \left. - \sqrt{X}\log\left[\frac{4X}{\mu^2}\right] - \sqrt{Y}\log\left[\frac{4Y}{\mu^2}\right] - \sqrt{Z}\log\left[\frac{4Z}{\mu^2}\right]\right) \\
& + \frac{1}{(16\pi^2)^2}(X+Y+Z)\left(-\log^2\left[\frac{e^{2\gamma}\mu^2}{16\pi^2 T^2}\right] - \log\left[\frac{e^{2\gamma}\mu^2}{16\pi^2 T^2}\right] + 2\gamma^2 + 4\gamma_1 - \frac{\pi^2}{4} - \frac{3}{2}\right) \\
& + \mathcal{O}(\tfrac{1}{T}).
\end{aligned}
\tag{2.7}
$$

Here $\gamma_1 \approx -0.0728$ is one of the Stieltjes constants: $\zeta(1+\epsilon) = \frac{1}{\epsilon} + \gamma - \gamma_1\epsilon + \mathcal{O}(\epsilon^2)$.

## 2.2 Fermionic sunset

The fermionic sunset is

$$
\begin{aligned}
\mathbf{I}_F(X,Y,Z) \equiv T^2 \sum_{n_p,n_q,n_l} \int_{\vec{p},\vec{q},\vec{l}} & \delta(\vec{p}+\vec{q}+\vec{l}) \delta_{n_p+n_q+n_l,0} \\
& \times \frac{1}{\vec{p}^2+X+(\pi(2n_p+1)T)^2} \frac{1}{\vec{q}^2+Y+(\pi(2n_q+1)T)^2} \frac{1}{\vec{l}^2+Z+(2\pi n_l T)^2} \\
& = \sum_{\{P,Q\}} \frac{1}{(P^2+X)(Q^2+Y)((P-Q)^2+Z)}.
\end{aligned}
\tag{2.8}
$$

Here propagators involving $P$ & $X$ and $Q$ & $Y$ are due to fermions—evidenced by the odd Matsubara frequencies.

Again, expand in $\epsilon$,

$$
\mathbf{I}_F(X,Y,Z) = \frac{(I_F)_{-2}(X,Y,Z)}{\epsilon^2} + \frac{(I_F)_{-1}(X,Y,Z)}{\epsilon} + I_F(X,Y,Z) + \mathcal{O}(\epsilon).
\tag{2.9}
$$

Divergent pieces are known to all orders, and the finite piece is zero to $\mathcal{O}(T^2)$,

$$
(I_F)_{-2}(X,Y,Z) = -\frac{1}{(16\pi^2)^2} \frac{X+Y+Z}{2},
\tag{2.10}
$$

$$
(I_F)_{-1}(X,Y,Z) = \frac{1}{16\pi^2}\left(A_F(X)+A_F(Y)+A(Z)-\frac{1}{16\pi^2}\frac{X+Y+Z}{2}\right),
\tag{2.11}
$$

where $A(Z)$ and $A_F(X)$ are the finite parts of the bosonic and fermionic 1-loop bubble integrals; see equations (3.10) and (3.12).

The finite piece $I_F(X,Y,Z)$ is zero to order $T^2$ [6,9], and the $T^0$ pieces can be inferred from [5]. In section 5.1 we derive all terms with odd powers of $T$, and in section 5.2 we derive the $T^0$ pieces. In summary,

$$
\begin{aligned}
I_F(X,Y,Z) = & -\frac{T\sqrt{Z}}{64\pi^3}\left(\log\left[\frac{e^{2\gamma}\mu^4}{4\pi^2 Z T^2}\right]+2\right) \\
& + \frac{1}{(16\pi^2)^2}\left\{(X+Y)\left(-\log^2\left[\frac{e^{2\gamma}\mu^2}{\pi^2 T^2}\right]-\log\left[\frac{e^{2\gamma}\mu^2}{\pi^2 T^2}\right]+2\gamma^2+4\gamma_1-\frac{\pi^2}{4}-\frac{3}{2}+4\log^2 2\right)\right. \\
& \left. +Z\left(-\log^2\left[\frac{e^{2\gamma}\mu^2}{16\pi^2 T^2}\right]-\log\left[\frac{e^{2\gamma}\mu^2}{16\pi^2 T^2}\right]+2\gamma^2+4\gamma_1-\frac{\pi^2}{4}-\frac{3}{2}+8\log^2 2\right)\right\} \\
& + \mathcal{O}(\tfrac{1}{T}).
\end{aligned}
\tag{2.12}
$$

Note the different logarithms for a bosonic versus a fermionic mass.

Sections 3–5 give detailed derivations of the above results.

## 3 High Temperature Expansions

Temperature dependence arises as multiplicative factors from Feynman rules and from propagators. The latter is an involved sum over Matsubara frequencies. Our approach is to perform the

high-temperature expansions before the Matsubara sums—not after. In this section we discuss this approach and introduce useful labels for the derivation to come. We give examples of how the high-temperature expansion works at one and two loops.

## 3.1 Hard/Soft split

Thermal integrals are often evaluated by first getting rid of all Matsubara sums. One advantage of this approach is that vacuum and thermal contributions are clearly separated. Also, because there are no sums left, remaining integrals can be performed numerically in the absence of analytical results.

To derive the high-temperature expansion of the sunset, we'll instead do the opposite: expand in $T$ before doing the Matsubara sums—the high-temperature expansion gets dealt with immediately. The problem is reduced to sums over master integrals. Separate momenta into hard and soft [10]:

$$\text{Hard } \vec{p}^2 \sim T^2,$$
$$\text{Soft } \vec{p}^2 \sim X,\, Y,\, Z.$$

Where a hierarchy between $T$ and the masses is assumed: $T^2 \gg X,\, Y,\, Z$.

Consider the bosonic propagator. There are two cases for hard momenta. First, for a finite Matsubara mode,

$$\frac{1}{P^2 + X} = \frac{1}{P^2} - \frac{X}{P^4} + \dots \tag{3.1}$$

Second, for a Matsubara zero-mode,

$$\frac{1}{\vec{p}^2 + X} = \frac{1}{\vec{p}^2} - \frac{X}{\vec{p}^4} + \dots \tag{3.2}$$

Likewise for soft momenta with a finite Matsubara mode,

$$\frac{1}{P^2 + X} = \frac{1}{(2\pi n_p T)^2} - \frac{\vec{p}^2 + X}{(2\pi n_p T)^4} + \dots, \tag{3.3}$$

and with a Matsubara zero-mode

$$\frac{1}{P^2 + X} = \frac{1}{\vec{p}^2 + X}. \tag{3.4}$$

Similar relations hold for the fermionic propagator.

## 3.2 1-loop momentum split

Take the bosonic 1-loop bubble integral

$$\mathbf{A}(X) \equiv \sum_P \frac{1}{P^2 + X} = \frac{A_{-1}(X)}{\epsilon} + A(X) + \mathcal{O}(\epsilon). \tag{3.5}$$

The traditional method proceeds by summing over Matsubara modes [11],

$$\mathbf{A}(X) = \int \frac{d^{d+1}p}{(2\pi)^{d+1}} \frac{1}{p^2 + X} + \int \frac{d^d p}{(2\pi)^d} \frac{1}{\sqrt{p^2 + X}} \frac{1}{e^{\beta\sqrt{p^2+X}} - 1}, \tag{3.6}$$
$$d = 3 - 2\epsilon,\ \beta = T^{-1}.$$

This is convenient since vacuum and temperature parts are separated; the Bose factor isolates the thermal part. The 4D part is readily evaluated

$$\int \frac{d^{d+1}p}{(2\pi)^{d+1}} \frac{1}{p^2 + X} = -\frac{X}{16\pi^2 \epsilon} + \frac{X}{16\pi^2}\left(\log\left[\frac{X}{\mu^2}\right] - 1\right) + \mathcal{O}(\epsilon), \tag{3.7}$$

where $\mu$ is the $\overline{\text{MS}}$ scale. There are various ways to evaluate the temperature integral; one is to use $\frac{1}{e^x - 1} = \frac{1}{x} - \frac{1}{2} + 2\sum_{l=1}^{\infty} \frac{z}{z^2 + (2\pi l)^2}$ to expand the Bose factor. The result is

$$\int \frac{d^d p}{(2\pi)^d} \frac{1}{\sqrt{p^2 + X}} \frac{1}{e^{\beta\sqrt{p^2+X}} - 1} = \frac{T^2}{12} - \frac{T\sqrt{X}}{4\pi} - \frac{X}{16\pi^2}\left(\log\left[\frac{Xe^{2\gamma}}{16\pi^2 T^2}\right] - 1\right) + \mathcal{O}(T^{-2}). \tag{3.8}$$

Adding the vacuum and thermal parts together gives the full result

$$A_{-1}(X) = -\frac{X}{16\pi^2}, \tag{3.9}$$

$$A(X) = \frac{T^2}{12} - \frac{T\sqrt{X}}{4\pi} - \frac{X}{16\pi^2}\log\left[\frac{\mu^2 e^{2\gamma}}{16\pi^2 T^2}\right] + \mathcal{O}(\epsilon, T^{-2}). \tag{3.10}$$

Note that all $\epsilon$-poles come from the vacuum part; thermal contributions can not diverge in the UV at one loop. There is a corresponding result for the fermionic 1-loop bubble,

$$(A_F)_{-1}(X) = -\frac{X}{16\pi^2}, \tag{3.11}$$

$$(A_F)(X) = -\frac{T^2}{24} - \frac{X}{16\pi^2}\log\left[\frac{\mu^2 e^{2\gamma}}{\pi^2 T^2}\right] + \mathcal{O}(\epsilon, T^{-2}). \tag{3.12}$$

An alternative derivation uses the hard/soft split. The high-temperature expansion and the momentum integration are done before the sums. That is,

$$\mathbf{A}(X) = \sum_P \frac{1}{P^2 + X} = \mathbf{A}_H + \mathbf{A}_S, \tag{3.13}$$

where $H$ and $S$ stand for hard and soft momenta respectively.

Start with the hard contribution,

$$\mathbf{A}_H = \sum_P \left(\frac{1}{P^2} - \frac{X}{P^4} + \mathcal{O}\left(\frac{X^2}{P^6}\right)\right)$$

$$= \frac{T^2}{12} - X\left[-\frac{\log\left(\frac{4\pi T}{\mu}\right)}{8\pi^2} + \frac{1}{16\pi^2 \epsilon} + \frac{\gamma}{8\pi^2}\right] + \mathcal{O}(T^{-2}) + \mathcal{O}(\epsilon). \tag{3.14}$$

The soft contribution is

$$\mathbf{A}_S = T\sum_{n\neq 0}\int_p\left\{\frac{1}{(2\pi n T)^2} - \frac{X + \vec{p}^2}{(2\pi n T)^4} + \dots\right\} + T\int_p \frac{1}{\vec{p} + X}$$

$$= -\frac{T\sqrt{X}}{4\pi} + \mathcal{O}(\epsilon) \tag{3.15}$$

where all contributions from $n \neq 0$ modes vanish due to scaleless integrals. Only zero-mode terms survive when all momenta are soft (this stays true at higher loop orders). One can similarly apply the hard/soft-split to the fermionic bubble. Though in this case there is no zero-mode and hence no soft contribution.

## 3.3   Sunset momentum split

There are two momentum integrals for the sunset, and so a few more cases. First take all Matsubara modes to zero. We use the notation $F \equiv \oint_{P,Q,L} \frac{1}{P^2+X} \frac{1}{Q^2+Y} \frac{1}{L^2+Z} \delta(\vec{p}+\vec{q}+\vec{l})\,|_{n_p=n_q=n_l=0}$.

$$F = F_{HHH} + F_{HHS} + F_{SSS} + \text{permutations}, \tag{3.16}$$

where $F_{HHS}$ is defined so that $\vec{p}$, $\vec{q}$ are hard, and $\vec{l}$ is soft. There are no terms with two soft momenta due to momentum conservation. Furthermore, $F_{HHH} = F_{HHS} = 0$ to all orders. For example, take $\vec{l}$ soft,

$$
\begin{aligned}
F_{HHS} &= T^2 \int_{\vec{p},\vec{l}} \frac{1}{(\vec{p}^2+X)\big((\vec{p}+\vec{l})^2+Y\big)(\vec{l}^2+Z)} \\
&= T^2 \int_{\vec{p},\vec{l}} \frac{1}{\vec{p}^4(\vec{l}^2+Z)} - T^2 \int_{\vec{p},\vec{l}} \big(X+Y+\vec{l}^2+4(\vec{p}\cdot\vec{l})^2\big)\frac{1}{\vec{p}^6}\frac{1}{\vec{l}^2+Z} + \ldots = 0, \tag{3.17}
\end{aligned}
$$

since all terms multiply a scaleless integral. Only the all-soft contribution is finite.

Next consider one zero and two finite modes, say $n_p = n_q \neq 0$, $n_l = 0$. Denote these as $G^l \equiv \oint_{P,Q,L} \frac{1}{P^2+X} \frac{1}{Q^2+Y} \frac{1}{L^2+Z} \delta(\vec{p}+\vec{q}+\vec{l})\,|_{n_p=n_q,n_l=0}$.

Again split the integral into different momentum regions,

$$G^l = G^l_{HHH} + G^l_{HHS} + G^l_{SSS} + \text{permutations}. \tag{3.18}$$

In this case only $G^l_{SSS}$ vanishes to all orders.

The leading order (in $T$) comes from $G^l_{HHS}$ and is of order $T$. Explicitly,

$$
\begin{aligned}
G^l_{HHS} &= T^2 \sum_{n_p} \int_{\vec{p},\vec{q},\vec{l}} \frac{1}{(P^2)(Q^2|_{n_q=n_p})(\vec{l}^2+Z)} \delta(\vec{p}+\vec{q}+\vec{l}) + \mathcal{O}\left(\sqrt{Z}\frac{X,\,Y,\,Z}{T}\right) \\
&= T\frac{Z^{\frac{1}{2}-\epsilon}}{64\pi^4}\left(\frac{e^\gamma \mu^2}{2\pi T}\right)^{2\epsilon}\Gamma\left[\epsilon-\frac{1}{2}\right]\Gamma\left[\epsilon+\frac{1}{2}\right]\zeta(1+2\epsilon) + \mathcal{O}\left(\sqrt{Z}\frac{X,\,Y,\,Z}{T}\right) \tag{3.19} \\
&= T\frac{\sqrt{Z}}{64\pi^3}\left(\log\left(\frac{64\pi^2 T^2 Z e^{-2\gamma}}{\mu^4}\right)-2-\frac{1}{\epsilon}\right) + \mathcal{O}(\epsilon) + \mathcal{O}\left(\sqrt{Z}\frac{X,\,Y,\,Z}{T}\right), \tag{3.20}
\end{aligned}
$$

other hard/soft momenta assignments of $G^l_{HHS}$ are zero to all orders of $T$; $G^l_{HHS}$ is given to all orders in the next section.

Finally there's the case when all Matsubara modes are finite. We use the notation $D \equiv \oint_{P,Q,L} \frac{1}{P^2+X} \frac{1}{Q^2+Y} \frac{1}{L^2+Z} \delta(\vec{p}+\vec{q}+\vec{l})\,|_{n_p\neq0,n_q\neq0,n_l\neq0}$. The momentum split is

$$D = D_{HHH} + D_{HHS} + D_{SSS} + \text{permutations}. \tag{3.21}$$

Note that $D_{HHS} = 0$,[1] and $D_{SSS} = 0$ to all orders. It is well known that

$$\sum_{P,Q} \oint \frac{1}{P^2}\frac{1}{Q^2}\frac{1}{L^2} = 0. \tag{3.22}$$

In the momentum split this statement means—to leading order in $T$—that $F_{HHH} + 3G^l_{HHH} + D_{HHH} = 0$, which is confirmed in the next section.

---

[1]This is a bit subtle, and care must be taken when evaluating the overall momentum delta function. For example, if we collapse the delta function as $\vec{q} = -\vec{p}-\vec{l}$, then all integrals are scaleless since $\vec{l}$ is soft. If the delta function is collapsed the other way, $\vec{l} = -\vec{p}-\vec{q}$, it seems like the result is finite. This is however only an illusion since for $\vec{l}$ to be soft we must have $\vec{p} = -\vec{q}+\vec{k}$, where $\vec{k}$ is soft. And all integrals vanish with this change of variables.

# 4 The Bosonic Sunset

There are two families of terms corresponding to odd and even powers of $T$. The former come from $G_{HHS}^l$ type integrals; the latter from $F_{SSS}$, $G_{HHH}^l$, $D_{HHH}$ and the permutations $G_{HHH}^q$, $G_{HHH}^p$.

The family of terms with odd powers of $T$ are always non-analytic in one of the masses-squared—we derive these terms in section 4.1. The $T^2$ terms are non-analytic while the remaining terms with even powers of $T$ are analytic.

The analytic terms can be written as

$$\mathbf{I}(X,Y,Z)\,|_{\text{analytic}} = \sum_{i,j,k\geq 0}(-X)^i(-Y)^j(-Z)^k \sum_{P,Q}\frac{1}{[P^2]^{(i+1)}}\frac{1}{[Q^2]^{(j+1)}}\frac{1}{[(P-Q)^2]^{(k+1)}}$$

$$\equiv \sum_{i,j,k\geq 0}(-X)^i(-Y)^j(-Z)^k L^d(i+1,j+1,k+1;00).$$

The function $L^d(i,j,k;00)$ corresponds to massless 2-loop integrals that can be calculated in terms of 1-loop functions by the use of IBP relations [12]. There is an algorithm for performing this reduction, and the master integrals required for the sunset $T^0$ term have been calculated in [4, 12]. In section 4.2 we give an independent derivation of these results.

## 4.1 Odd powers of $T$

The starting point is

$$G_{HHS}^l = T^2\sum_{n\neq 0}\int_{\vec{p},\vec{l}}\frac{1}{(\vec{p}^2+(2\pi nT)^2+X)((\vec{p}+\vec{l})^2+(2\pi nT)^2+Y)(\vec{l}^2+Z)},\qquad(4.1)$$

where $\vec{p}\sim\vec{q}\sim T$, and $\vec{l}^2\sim Z\ll T^2$. There are two sources of higher-order terms. The first type comes from using the delta function and expanding $(p+l)^2$ in powers of $l^2$ (odd $l$ terms integrate to zero). These terms give corrections of order $\frac{Z}{T^2}$. Remaining terms come from expanding the finite-mode propagators in powers of $X$ and $Y$. Ignoring $\mathcal{O}(\epsilon)$ corrections, the general result is

$$G_{HHS}^l = \mathcal{O}(T) - \frac{\sqrt{Z}}{4\pi}\sum_{\alpha=1}^{\infty}(2\pi)^{-2\alpha-3}\Gamma\left(\alpha+\frac{1}{2}\right)T^{1-2\alpha}\zeta(2\alpha+1)(-1)^\alpha$$

$$\times\sum_{m=0}^{2\alpha}\sum_{i=\max(0,m-\alpha)}^{\lfloor\frac{m}{2}\rfloor}4^i\frac{\binom{m}{2i}\Gamma\left[i+\frac{1}{2}\right]}{\Gamma[i+\alpha+2]}Z^i(Y-Z)^{m-2i}X^{i+\alpha-m},\qquad(4.2)$$

where $\alpha=0$ is a special case, due to an $\epsilon$-pole, and must be treated separately—see equation (3.20). Note that $G_{HHS}^l$ is symmetric in $X$ and $Y$ as it must be. For example, the $T^{-1}$ and $T^{-3}$ contributions are respectively

$$G_{HHS}^l = \mathcal{O}(T) + 2T^{-1}\sqrt{Z}\frac{\zeta(3)(3(X+Y)-Z)}{3(4\pi)^5}$$

$$- 2T^{-3}\sqrt{Z}\frac{\zeta(5)\left(10\left(X^2+XY+Y^2\right)-5Z(X+Y)+Z^2\right)}{5(4\pi)^7} + \mathcal{O}\left(T^{-5}\right).\qquad(4.3)$$

## 4.2 Even powers of $T$

The momentum integrals are more clear after Fourier transforming to coordinate space [13],

$$V(R,m) = \int_{\vec{k}} e^{i\vec{k}\cdot\vec{R}} \frac{1}{\vec{k}^2 + m^2} = \left(\frac{e^\gamma \mu^2}{4\pi}\right)^\epsilon \frac{1}{(2\pi)^{3/2-\epsilon}} \left(\frac{m}{R}\right)^{1/2-\epsilon} K_{1/2-\epsilon}(mR). \qquad (4.4)$$

The coordinate space representation is then

$$\frac{1}{\vec{k}^2 + m^2} = \int_R V(R,m) e^{-i\vec{k}\cdot\vec{R}}, \qquad (4.5)$$

with measure $\int_R \equiv \left(\frac{e^\gamma \mu^2}{4\pi}\right)^{-\epsilon} \int d^{3-2\epsilon} R$. Performing this replacement in the sunset eliminates all momentum integrals, leaving a single $R$ integral. The propagator's coordinate representation in any dimension is derived in appendix C.

### 4.2.1 Order $T^2$

There are *a priori* three different contributions at order $T^2$. The all zero-mode contribution $F_{SSS}$, and the two finite mode contributions $G^l_{HHH}, D_{HHH}$. Finite mode contributions do not depend on the masses and are zero. And so $F_{SSS}$ gives the full order $T^2$ result. The integral can be performed using standard techniques [14].

To show that the mass-independent $T^2$ term vanishes, start with $G^l_{HHH}$,

$$G^l_{HHH} = \sum_{P,Q} \frac{1}{P^2 Q^2} \frac{1}{(\vec{p}+\vec{q})^2} + \mathcal{O}\left(T^0\right). \qquad (4.6)$$

Ignore $T^0$ terms for now, $G^l_{HHH}$ simply refers to $T^2$ terms in this section. From now on, unless otherwise specified, Matsubara modes $n$, $m$, and $l$ always refer to the absolute value of said integer, and are all positive. Transforming to coordinate space we find

$$\begin{aligned}
G^l_{HHH} &= T^2 \sum_{n\neq 0} \int_0^\infty dR \frac{2^{-2\epsilon-3} e^{2\gamma\epsilon} \mu^{4\epsilon}}{\pi^3 (1-2\epsilon)} \left[R^\epsilon K_{\epsilon-\frac{1}{2}}(2\pi T n R)\right]^2 (2\pi n T)^{1-2\epsilon} \\
&= -T^2 \frac{2^{-2\epsilon-4} e^{2\gamma\epsilon} \zeta(4\epsilon) \left(\frac{\mu}{2\pi T}\right)^{4\epsilon} \Gamma\left(\epsilon+\frac{1}{2}\right)\Gamma(2\epsilon-1)}{\pi^{5/2}\Gamma(\epsilon+1)}.
\end{aligned} \qquad (4.7)$$

The second integral is

$$D_{HHH} = T^2 \sum_{n,m,l\neq 0} \int_{\vec{p},\vec{q}} \frac{1}{(\vec{p}^2 + (2\pi n T)^2)} \frac{1}{(\vec{q}^2 + (2\pi m T)^2)} \frac{1}{((\vec{p}+\vec{q})^2 + (2\pi l T)^2)} \delta_{n+m+l,0}. \qquad (4.8)$$

Rescaling the momenta and going to coordinate space,

$$\begin{aligned}
D_{HHH} = T^2 \sum_{n,m,l\neq 0} &\frac{2^{-\epsilon-\frac{7}{2}} e^{2\gamma\epsilon}}{\pi^3 \Gamma\left(\frac{3}{2}-\epsilon\right)} \left(\frac{\mu}{2\pi T}\right)^{4\epsilon} (lmn)^{\frac{1}{2}-\epsilon} \delta_{n+m+l,0} \times \\
&\int_0^\infty dR\, R^{\epsilon+\frac{1}{2}} K_{\epsilon-\frac{1}{2}}(lR) K_{\epsilon-\frac{1}{2}}(mR) K_{\epsilon-\frac{1}{2}}(nR).
\end{aligned} \qquad (4.9)$$

The integral is known in closed form [15, 16],

$$
\int_0^\infty dR\, R^{\epsilon+\frac{1}{2}} K_{\epsilon-\frac{1}{2}}(lR) K_{\epsilon-\frac{1}{2}}(mR) K_{\epsilon-\frac{1}{2}}(nR) \tag{4.10}
$$

$$
= -\left(\frac{\pi}{2}\right)^{5/2} 2^{\nu-1/2} \frac{\Gamma(1/2-\nu)}{\pi \sin \pi \nu} \frac{\Delta^{2\nu-1}}{(nlm)^\nu}
$$

$$
+ \frac{\pi}{\sin \pi \nu} 2^{2\nu-3} \sqrt{\frac{\pi}{2}} \Gamma(1-2\nu) \frac{\Delta^{2\nu-1}}{(nlm)^\nu} \left\{ (\sin \phi_n)^{1/2-\nu} P_{\nu-1/2}^{\nu-1/2}(\cos \phi_n) + (n \to m, l) \right\},
$$

$$
\phi_n = \sec^{-1}\left(\frac{2lm}{l^2+m^2-n^2}\right),
$$

$$
\phi_m = \sec^{-1}\left(\frac{2ln}{l^2-m^2+n^2}\right)
$$

$$
\phi_l = \sec^{-1}\left(\frac{2mn}{-l^2+m^2+n^2}\right) + 2\pi
$$

$$
\Delta = \frac{1}{2} mn \sqrt{1 - \frac{(-l^2+m^2+n^2)^2}{4m^2n^2}}
$$

$$
\nu = 1/2 - \epsilon.
$$

Yet there is a subtlety. The formula is technically not valid when $l = n + m$, which is precisely the case of interest. But the formula holds in dimensional regularisation. The trick is to enforce the Kronecker delta by setting $l = n + m - \delta^2$ ($l$ is here the absolute value), and then take the $\delta \to 0$ limit. All $\delta$ dependence cancels to $\mathcal{O}(\delta^2)$, so the $\delta \to 0$ limit is valid.

When all is said and done

$$
D_{HHH} = -T^2 \sum_{n,m\neq 0} \frac{2^{-2\epsilon-5} e^{2\gamma\epsilon} \left(\frac{\mu}{2\pi T}\right)^{4\epsilon} \csc(2\pi\epsilon) \Gamma\left(\epsilon+\frac{1}{2}\right)\left(-m^{2\epsilon}+(m+n)^{2\epsilon}-n^{2\epsilon}\right)}{\pi^{3/2}\Gamma(2-2\epsilon)\Gamma(\epsilon+1)(mn(m+n))^{2\epsilon}}. \tag{4.11}
$$

So adding the two contributions $D_{HHH} + G^l_{HHH}$ gives

$$
D_{HHH} + G^l_{HHH} + \text{permutations} = -T^2 \left(\frac{\mu}{2\pi T}\right)^{4\epsilon} \frac{2^{-2\epsilon-5} e^{2\gamma\epsilon} \csc(2\pi\epsilon)\Gamma\left(\epsilon+\frac{1}{2}\right)}{\pi^{3/2}\Gamma(2-2\epsilon)\Gamma(\epsilon+1)} \times
$$

$$
\left\{ 6 \times \sum_{n,m\neq 0} \left[ \frac{-m^{2\epsilon}+(m+n)^{2\epsilon}-n^{2\epsilon}}{(mn(m+n))^{2\epsilon}} \right] - 3 \times 2\zeta(4\epsilon) \right\}. \tag{4.12}
$$

The numerical factors above come from different summation regions in the first case, and permutations of the zero-mode in the second. There are 6 summation regions in total:

$$
(i): \ l < 0 \ m, n > 0 \qquad l > 0 \ m, n < 0, \tag{4.13}
$$

$$
(ii): \ n < 0 \ l, m > 0 \qquad n > 0 \ l, m < 0, \tag{4.14}
$$

$$
(iii): \ m < 0 \ n, l > 0 \qquad m > 0 \ n, l < 0. \tag{4.15}
$$

Contributions from $l < 0, \ m, n > 0$ is the same as $l > 0, \ m, n < 0$ since the sums only involve the absolute values of $l, \ m, \ n$. So take region $(i)$. In this region we collapse the delta function on $l$ and set $l = m + n$; the sum is then over $n, m = 1, 2, \ldots$. Similarly for $(ii)$: collapse the delta function on $n$ and set $n = l + m$. But our expressions are symmetric in $n, m, l$ so all sums are the same. This gives a factor of 6.

Non-trivial sums are

$$\sum_{n,m=1}^{\infty}\left[\frac{-m^{2\epsilon}+(m+n)^{2\epsilon}-n^{2\epsilon}}{(mn(m+n))^{2\epsilon}}\right]=\zeta(2\epsilon)^2-2\sum_{n,m=1}^{\infty}\frac{1}{n^{2\epsilon}(n+m)^{2\epsilon}}. \tag{4.16}$$

Using the methods in appendix A the second sum is

$$\sum_{n,m=1}^{\infty}\frac{1}{n^{2\epsilon}(n+m)^{2\epsilon}}=\frac{1}{2}\left[\zeta(2\epsilon)^2-\zeta(4\epsilon)\right]. \tag{4.17}$$

Add everything together to find

$$-6\times T^2\left(\frac{\mu}{2\pi T}\right)^{4\epsilon}\frac{2^{-2\epsilon-5}e^{2\gamma\epsilon}\csc(2\pi\epsilon)\Gamma\left(\epsilon+\frac{1}{2}\right)}{\pi^{3/2}\Gamma(2-2\epsilon)\Gamma(\epsilon+1)}\times$$
$$\left\{\zeta(2\epsilon)^2-2\frac{1}{2}\left(\zeta(2\epsilon)^2-\zeta(4\epsilon)\right)-\zeta(4\epsilon)\right\}=0. \tag{4.18}$$

So the mass independent $T^2$ term vanishes as promised.

### 4.2.2 Order $T^0$

Order $T^0$ gets contributions from two terms: $G_{HHH}^{l}$ and $D_{HHH}$. Again, start with $G_{HHH}^{l}$. The relevant terms are

$$G_{HHH}^{l}=\mathcal{O}\left(T^2\right)+\sum_{P,Q}\frac{1}{P^2Q^2(\vec{p}+\vec{q})^2}\left[-\frac{X+Y}{P^2}-\frac{Z}{(\vec{p}+\vec{q})^2}\right]+\mathcal{O}\left(T^{-2}\right). \tag{4.19}$$

For clarity $G_{HHH}^{l}$ refers to the $T^0$ term for the remainder of this section.

The trick to evaluating these integrals in coordinate space is to rewrite powers of the propagators as derivatives acting on it, as in

$$\frac{1}{(\vec{p}^2+m^2)^2}=-\frac{1}{2m}\partial_m\frac{1}{\vec{p}^2+m^2}. \tag{4.20}$$

This trick can be used to calculate

$$G_{HHH}^{l}\left(\frac{\mu}{2\pi T}\right)^{-4\epsilon}\zeta(2+4\epsilon)^{-1}=\frac{X+Y}{(2\pi)^2}\frac{e^{2\gamma\epsilon}(2\epsilon-1)\Gamma\left(\epsilon-\frac{1}{2}\right)^2}{128\pi^3}$$
$$-\frac{Z}{(2\pi)^2}\frac{2^{-2\epsilon-7}e^{2\gamma\epsilon}\Gamma\left(\epsilon-\frac{1}{2}\right)\Gamma(2\epsilon+1)}{\pi^{5/2}\Gamma(\epsilon+2)}. \tag{4.21}$$

There are also contributions from permutations of the zero-mode. Adding them together and expanding in $\epsilon$ gives

$$G_{HHH}^{p}+G_{HHH}^{q}+G_{HHH}^{l}=-3\zeta(2)\frac{(X+Y+Z)}{(4\pi)^4}+\mathcal{O}(\epsilon). \tag{4.22}$$

Note that there are no $\epsilon$-divergences. This is not surprising. Before the sum, by dimensional reasons, the $n$ power must be $n^{-4\epsilon-2}$.[2] This can never diverge; similar for higher-order terms.

---

[2]Because Matsubara modes only appear in the combination $nT$.

The situation is different however when three modes are finite. On dimensional grounds the sum must be of the form

$$\sum_{n,l,m} \frac{1}{n^a m^b (n+m)^c},$$ 

(4.23)

where $a+b+c = 4\epsilon+2$. Yet there are now many possible divergent combinations. For example, taking $a = b = \epsilon+1$, $c = 2\epsilon$ gives double and single poles in $\epsilon$. These divergences are similar to the standard overlapping divergences at zero temperature.

Next is the contribution from $D_{HHH}$. Start with terms proportional to $Z$—the others are obtained by symmetry. The integral is

$$D_{HHH} = \mathcal{O}\left(T^2\right) - Z \sum_{P,Q} \frac{1}{P^4 Q^2 (P+Q)^2} + \mathcal{O}\left(T^{-2}\right).$$ 

(4.24)

Mimicking $G^l_{HHH}$, $D_{HHH}$ only denotes the $T^0$ term. In coordinate space the integral is

$$-\sum_{P,Q} \frac{1}{P^4 Q^2 (P+Q)^2} = \sum_{n,m,l \neq 0} A\, \delta_{n+m+l,0} \int_0^\infty dR\, R^{\epsilon+3/2} K_{\epsilon-\frac{1}{2}}(lR) K_{\epsilon-\frac{1}{2}}(mR) K_{\epsilon+\frac{1}{2}}(nR),$$ 

(4.25)

$$A = -\frac{2^{-\epsilon-\frac{9}{2}} e^{2\gamma\epsilon} (lmn)^{1/2-\epsilon}}{n\pi^3 \Gamma\left(\frac{3}{2}-\epsilon\right)} (2\pi)^{-2} \left(\frac{\mu}{2\pi T}\right)^{4\epsilon}.$$ 

(4.26)

The integral can again be done in closed form. Rewrite the squared propagator using the trick above, and then use the relation [15, 16]

$$\int_0^\infty dR\, R^{\epsilon+3/2} K_{\epsilon-\frac{1}{2}}(lR) K_{\epsilon-\frac{1}{2}}(mR) K_{\epsilon+\frac{1}{2}}(nR)$$
$$= -\left(\frac{d}{dn} + \frac{1/2-\epsilon}{n}\right) \int_0^\infty dR\, R^{\epsilon+\frac{1}{2}} K_{\epsilon-\frac{1}{2}}(lR) K_{\epsilon-\frac{1}{2}}(mR) K_{\epsilon-\frac{1}{2}}(nR),$$ 

(4.27)

where the second integral is given by equation (4.10). Again we imagine collapsing the Kronecker delta by setting $l = n+m-\delta^2$, and taking the $\delta \to 0$ limit. There are some new subtleties because of $\delta^{-2}$ terms, but these all cancel out, and the $\delta \to 0$ limit can be consistently taken.

After some simplifications one finds

$$A \int_0^\infty dR\, R^{\epsilon+3/2} K_{\epsilon-\frac{1}{2}}(lR) K_{\epsilon-\frac{1}{2}}(mR) K_{\epsilon+\frac{1}{2}}(nR) \mid_{l=m+n}$$
$$= B \frac{1}{n[mn(m+n)]^{2\epsilon+1}} \Big\{ 2n(\epsilon+1)m^{2\epsilon+1} + (2\epsilon+1)m^{2\epsilon+2} - m^2(2\epsilon+1)(m+n)^{2\epsilon}$$
$$+ n^2\left((m+n)^{2\epsilon} - n^{2\epsilon}\right) - 2mn\epsilon(m+n)^{2\epsilon} \Big\},$$ 

(4.28)

$$B = -\frac{2^{-2(\epsilon+4)} e^{2\gamma\epsilon} \sec(\pi\epsilon) \Gamma(2\epsilon+1)}{\pi^{3/2} \Gamma\left(\frac{3}{2}-\epsilon\right) \Gamma(\epsilon+2)} (2\pi)^{-2} \left(\frac{\mu}{2\pi T}\right)^{4\epsilon}.$$ 

(4.29)

This result can be double-checked order-by-order in $\epsilon$ by expanding the Bessel functions before integrating. Using this we have double-checked all integrals to $\mathcal{O}(\epsilon)$.

Again, same as for the $T^2$ term, there are 6 summation regions.

$$(i): \ l < 0, \ m, n > 0 \qquad l > 0, \ m, n < 0,$$ 

(4.30)

$$(ii): \ n < 0, \ l, m > 0 \qquad n > 0, \ l, m < 0,$$ 

(4.31)

$$(iii): \ m < 0, \ n, l > 0 \qquad m > 0, \ n, l < 0.$$ 

(4.32)

All flipped regions are symmetric and gives an overall factor of 2. So it is enough to consider the left column. Using the convention that we always collapse the Kronecker delta on the negative mode gives

$$(i): \ l = m + n, \tag{4.33}$$

$$(ii): \ n = l + m, \tag{4.34}$$

$$(iii): \ m = l + n. \tag{4.35}$$

Note that region $(iii)$ gives the same contribution as $(i)$ since they are related by relabelling dummy indices. Region $(ii)$ is different—because the negative mode is in the $P^{-4}$ propagator. We have to do this case separately.

So yet another integral is needed:

$$
- T^2 \sum_{n,m=1}^{\infty} \int_{\vec{p},\vec{q}} \frac{1}{P_{n+m}^4 Q_n^2 (P+Q)_m^2}
$$
$$
= B \sum_{n,m=1}^{\infty} m^{-2\epsilon-1} n^{-2\epsilon-1} (m+n)^{-2(\epsilon+1)} \left\{ 2n(\epsilon+1)m^{2\epsilon+1} + m^{2\epsilon+2} - m^2(m+n)^{2\epsilon} \right.
$$
$$
\left. + 2mn\left((\epsilon+1)n^{2\epsilon} - (m+n)^{2\epsilon}\right) + n^2\left(n^{2\epsilon} - (m+n)^{2\epsilon}\right) \right\}. \tag{4.36}
$$

With the same $B$ as in equation (4.28). After doing the sums we get the neat result

$$
- \sum\!\!\!\!\!\!\int_{P,Q} \frac{1}{P^4 Q^2 (P+Q)^2} = B \times 4\left[ \zeta(2\epsilon+1)^2 - (2\epsilon + \frac{3}{2})\zeta(4\epsilon+2) \right]. \tag{4.37}
$$

The result of adding $D_{HHH}$ and $G_{HHH}^l$ + permutations is given in (5.26); or after expanding to $\mathcal{O}(\epsilon^0)$,

$$
(4\pi)^4 (X+Y+Z)^{-1} (D_{HHH} + G_{HHH}^l + \ldots) = \left(\frac{\mu}{2\pi T}\right)^{4\epsilon} \left\{ -\frac{1}{2\epsilon^2} - \frac{1+4\gamma - 4\log 2}{2\epsilon} \right.
$$
$$
\left. + 4\gamma_1 + \frac{\pi^2}{4} - \frac{3}{2} - 4\log^2(2) - 2\gamma(\gamma + 1 - 4\log(2)) + \log(4) - 3\zeta(2) \right\}
$$
$$
= -\frac{1}{2\epsilon^2} + \frac{1}{\epsilon}\left[ 2\log\left(\frac{4\pi T}{\mu}\right) - 2\gamma - \frac{1}{2} \right]
$$
$$
+ 2(1+4\gamma)\log\left(\frac{4\pi T}{\mu}\right) - 4\log^2\left(\frac{4\pi T}{\mu}\right) + 4\gamma_1 - 2\gamma^2 + \frac{\pi^2}{4} - \frac{3}{2} - 2\gamma - 3\zeta(2). \tag{4.38}
$$

So all $\epsilon$ poles come from $D_{HHH}$. The $\epsilon$ poles above agree with previous results [17], and the finite pieces agree with [4].

## 5 The Fermionic Sunset

The fermionic sunset's high-temperature expansion is similar to the bosonic. But now only the $l$ propagator can have a zero Matsubara mode; and so only the $l$ momentum can be soft. We also have to keep track of which Matsubara modes are odd or even.

We use the same notation $G, D$ as for the boson. In this case

$$G = G_{HHH}^l + G_{HHS}^l, \tag{5.1}$$

because $G^l_{HHS}$ is the only finite configuration with one soft momentum. All other contributions, including $l \to n,\, m$, vanish. That leaves us with

$$\mathbf{I}_F(X, Y, Z) = G^l_{HHH} + G^l_{HHS} + D_{HHH}. \tag{5.2}$$

Just as in the bosonic case, we derive all the non-analytic terms coming with odd powers of $T$, in section 5.1. For the even powers of $T$ all terms are analytic in the squared masses, and we can write them as

$$\mathbf{I}_F(X, Y, Z)\,|_{\text{analytic}} = \sum_{i,j,k \geq 0} (-X)^i (-Y)^j (-Z)^k \oint_{\{P,Q\}} \frac{1}{[P^2]^{(i+1)}} \frac{1}{[Q^2]^{(j+1)}} \frac{1}{[(P-Q)^2]^{(k+1)}}$$

$$\equiv \sum_{i,j,k \geq 0} (-X)^i (-Y)^j (-Z)^k \widehat{L}^d(i+1, j+1; k+1; 00).$$

Here we defined the function $\widehat{L}^d(i, j; k; 00)$, which is the fermionic counterpart to $L^d(i, j, k; 00)$ and can similarly be reduced to 1-loop functions by the use of IBP relations. For the $T^0$ term, the relevant subset can be derived from [5].[3] In section 5.2 we give an independent derivation of these results.

## 5.1  Odd powers of $T$

Odd $T$ powers are calculated analogously to the bosonic sunset. All arise from the integral

$$G^l_{HHS} = T^2 \sum_{n \neq 0} \int_{\vec{p},\vec{l}} \frac{1}{(\vec{p}^2 + [\pi(2n+1)T]^2 + X)((\vec{p}+\vec{l})^2 + [\pi(2n+1)T]^2 + Y)(\vec{l}^2 + Z)}. \tag{5.3}$$

The $\mathcal{O}(T)$ contribution is

$$G^l_{HHS} = \frac{T Z^{\frac{1}{2}-\epsilon}}{64\pi^4} 2^{-2\epsilon} \left(2^{1+2\epsilon} - 1\right) \left(\frac{e^\gamma \mu^2}{\pi^2 T^2}\right)^{2\epsilon} \Gamma\!\left[\epsilon - \frac{1}{2}\right] \Gamma\!\left[\epsilon + \frac{1}{2}\right] \zeta(2\epsilon+1) + \mathcal{O}(T^{-1}). \tag{5.4}$$

Ignoring $\mathcal{O}(\epsilon)$ corrections, the general result for higher orders is

$$G^l_{HHS} = \mathcal{O}(T) - \frac{\sqrt{Z}}{4\pi} \sum_{\alpha=1}^{\infty} (2^{2\alpha+1} - 1)(2\pi)^{-2\alpha-3} \Gamma\!\left(\alpha + \frac{1}{2}\right) T^{1-2\alpha} \zeta(2\alpha+1)$$

$$\times \sum_{m=0}^{2\alpha} \sum_{i=\max(0,m-\alpha)}^{\lfloor \frac{m}{2} \rfloor} 4^i \frac{\binom{m}{2i} \Gamma\!\left[i + \frac{1}{2}\right]}{\Gamma[i + \alpha + 2]}$$

$$\times (-1)^\alpha Z^i (Y - Z)^{m-2i} X^{i+\alpha-m}. \tag{5.5}$$

## 5.2  Even powers of $T$

### 5.2.1  Order $T^2$

It's not necessary to perform any calculations for the $T^2$ contribution because it can be shown to be zero by the following argument. First, two Matsubara modes are always finite, so the $T^2$ contribution can't depend on masses. Second, the remaining contribution is of the form

$$\oint_{\{P,Q\}} \frac{1}{P^2 Q^2 L^2}. \tag{5.6}$$

---

[3]Here we are not using the same notation as [5]. Our $\widehat{L}^d(i, j; k; 00)$ corresponds to their $Z_{ijk}$.

And we can use the summation trick in [6] to reshuffle things as

$$\sum_{\{P,Q\}} \frac{1}{P^2 Q^2 L^2} = \frac{2^{4\epsilon}-1}{3} \sum_{P,Q} \frac{1}{P^2 Q^2 L^2}. \tag{5.7}$$

But the right-hand side is zero—so the fermionic sunset is zero at $\mathcal{O}\left(T^2\right)$.

Nevertheless—since it introduces methods we will need later—let's explicitly show that the $T^2$ contribution vanishes to all orders in $\epsilon$.

There are two contributions at order $T^2$. First,

$$G^l_{HHH} = -2T^2 \frac{2^{-2\epsilon-5} e^{2\gamma\epsilon} \Gamma\left(\epsilon+\frac{1}{2}\right) \Gamma(2\epsilon-1) \left(\frac{\mu}{\pi T}\right)^{4\epsilon}}{\pi^{5/2} \Gamma(\epsilon+1)} (1-2^{-4\epsilon})\zeta(4\epsilon), \tag{5.8}$$

and second, $D_{HHH}$:

$$D_{HHH} = -T^2 \sum \frac{2^{-2\epsilon-5} e^{2\gamma\epsilon} \left(\frac{\mu}{\pi T}\right)^{4\epsilon} \csc(2\pi\epsilon) \Gamma\left(\epsilon+\frac{1}{2}\right) \left(-m^{2\epsilon}+(m+n)^{2\epsilon}-n^{2\epsilon}\right)}{\pi^{3/2} \Gamma(2-2\epsilon) \Gamma(\epsilon+1)(mn(m+n))^{2\epsilon}}. \tag{5.9}$$

In this case two modes are fermionic, and one is bosonic. Without loss of generality we'll take $n$ as the bosonic mode. There are six summation regions:

$$(i): \ l<0, \ m,n>0 \qquad l>0, \ m,n<0, \tag{5.10}$$

$$(ii): \ n<0, \ l,m>0 \qquad n>0, \ l,m<0, \tag{5.11}$$

$$(iii): \ m<0, \ n,l>0 \qquad m>0, \ n,l<0. \tag{5.12}$$

Again, the convention is that the Kronecker delta collapses upon the negative mode. So in region (i) $l$ is odd and is $l=m+n$; $m$ is odd and $n$ is even. All flipped regions give an overall factor of two and region $(i)$ and $(iii)$ are identical.

Up to an irrelevant prefactor, the result is

$$D_{HHH} \propto \left[ 2\left( 2^{-2\epsilon}(1-2^{-2\epsilon})\zeta(2\epsilon)^2 - \left\{\sum_{eo}+\sum_{oo}\right\} \frac{1}{n^{2\epsilon}(n+m)^{2\epsilon}} \right) \right.$$
$$\left. +(1-2^{-2\epsilon})^2\zeta(2\epsilon)^2 - 2\sum_{oe} \frac{1}{n^{2\epsilon}(n+m)^{2\epsilon}} \right], \tag{5.13}$$

where $\sum_{eo} \frac{1}{n^a(n+m)^b}$ stands for the sum over (positive) even $n$ and odd $m$. Adding $G^l_{HHH}$ gives

$$D_{HHH} + G^l_{HHH} \propto \left[ 2\left( 2^{-2\epsilon}(1-2^{-2\epsilon})\zeta(2\epsilon)^2 - \left\{\sum_{eo}+\sum_{oo}\right\} \frac{1}{n^{2\epsilon}(n+m)^{2\epsilon}} \right) \right.$$
$$\left. +(1-2^{-2\epsilon})^2\zeta(2\epsilon)^2 - 2\sum_{oe} \frac{1}{n^{2\epsilon}(n+m)^{2\epsilon}} - (1-2^{-4\epsilon})\zeta(4\epsilon) \right]. \tag{5.14}$$

This vanishes after using identities from appendix A. So the fermionic sunset is indeed zero at order $T^2$.

### 5.2.2 Order $T^0$

All contributions at order $T^0$ come from $G^l_{HHH} + D_{HHH}$. First, note that if all squared masses are identical $(= X)$,

$$G^l_{HHH} + D_{HHH} = X(2^{2+4\epsilon}-1) \sum_{P,Q} \frac{1}{(P^4)(Q^2)(L^2)} \delta(\vec{p}+\vec{q}+\vec{l}), \tag{5.15}$$

where all modes on the right-hand side are bosonic. Or written differently $\mathbf{I}_F(X,X,X) = (2^{2+4\epsilon} - 1)\mathbf{I}(X,0,0)$ at order $T^0$—similar relations holds for other $T$ orders. This provides a convenient cross-check.

Starting with $D_{HHH}$—where prefactors are left implicit and orders other than $T^0$ are dropped—we find for the bosonic mass term $D_{HHH}^Z$,

$$
\begin{aligned}
D_{HHH}^Z \propto 2\Bigg\{ &(2\epsilon+1)\Bigg[ \sum_{eo} n^{-2\epsilon-2}(n+m)^{-2\epsilon} \\
&+ \sum_{oo} n^{-2\epsilon}(n+m)^{-2\epsilon-2} - (1-2^{-2\epsilon})2^{-2\epsilon-2}\zeta(2\epsilon+2)\zeta(2\epsilon)\Bigg] \\
&+ \Bigg(\sum_{eo}+\sum_{oo}\Bigg)\frac{1}{n^{2\epsilon+1}(n+m)^{2\epsilon+1}} - \sum_{oe}\frac{1}{n^{2\epsilon+1}(n+m)^{2\epsilon+1}} \\
&+(1-2^{-2\epsilon-1})2^{-2\epsilon-1}\zeta(2\epsilon+1)^2 - \frac{1}{2}(1-2^{-2\epsilon-1})^2\zeta(2\epsilon+1)^2\Bigg\},
\end{aligned}
\tag{5.16}
$$

where all contributions from region (i)-(iii) have been added. All sums can be done in closed form using various identities in appendix A. In terms of $\zeta$-functions the result is neatly expressed as

$$
D_{HHH}^Z = B_F \times 4^{-2\epsilon-1}\left(2^{2\epsilon+1}-1\right)\left[\left(4^{\epsilon+1}-6\right)\zeta(2\epsilon+1)^2 - \left(2^{2\epsilon+1}+1\right)\zeta(4\epsilon+2)\right],
\tag{5.17}
$$

$$
B_F = \frac{\pi^{3/2}4^{1-\epsilon}e^{2\gamma\epsilon}\epsilon\csc(\pi\epsilon)\sec^2(\pi\epsilon)}{\Gamma(2-2\epsilon)\Gamma\left(\frac{1}{2}-\epsilon\right)\Gamma(\epsilon+2)}\frac{1}{(4\pi)^4}\left(\frac{\mu}{2\pi T}\right)^{4\epsilon}2^{4\epsilon}.
\tag{5.18}
$$

So after expanding to $\mathcal{O}\left(\epsilon^0\right)$ the bosonic mass contribution is

$$
\begin{aligned}
D_{HHH} \supset Z\left(\frac{\mu}{\pi T}\right)^{4\epsilon}\frac{1}{(4\pi)^4}\Bigg[ &-\frac{1}{2\epsilon^2} - \frac{1+4\gamma-4\log 4}{2\epsilon} \\
&+4\gamma_1-2\gamma^2-\frac{3}{2}-\frac{3\pi^2}{4}-2\gamma-8\log^2(2)+16\gamma\log(2)+\log(16)\Bigg].
\end{aligned}
\tag{5.19}
$$

Let's continue with the contribution of a fermionic mass, $D_{HHH}^X$. Following the same steps as before, we find

$$
D_{HHH}^X = -B_F\left[2^{-4\epsilon-1}\left(\left(2^{2\epsilon+1}-1\right)^2\zeta(2\epsilon+1)^2 - \left(4^{2\epsilon+1}-1\right)(\epsilon+1)\zeta(4\epsilon+2)\right)\right],
\tag{5.20}
$$

with $B_F$ as in equation (5.18). Including both fermionic masses, the proper prefactors, and expanding to $\mathcal{O}\left(\epsilon^0\right)$,

$$
\begin{aligned}
D_{HHH} \supset (X+Y)\left(\frac{\mu}{\pi T}\right)^{4\epsilon}\frac{1}{(4\pi)^4}\Bigg\{ &-\frac{1}{2\epsilon^2} - \frac{1+4\gamma}{2\epsilon} \\
&+4\gamma_1-2\gamma^2+\frac{3\pi^2}{4}-\frac{3}{2}-2\gamma+4\log^2(2)\Bigg\}.
\end{aligned}
\tag{5.21}
$$

This completes the contribution from $D_{HHH}$.

Which only leaves $G_{HHH}^l$. From the bosonic mass term,

$$
\begin{aligned}
G_{HHH}^l \supset &-Z\left(\frac{\mu}{\pi T}\right)^{4\epsilon}\frac{e^{2\gamma\epsilon}\Gamma\left[\epsilon-\frac{1}{2}\right]\Gamma\left[2\epsilon+1\right]}{128\pi^{9/2}\Gamma\left[\epsilon+2\right]}4^{-\epsilon}(1-2^{-4\epsilon-2})\zeta(4\epsilon+2) \\
&= 3\frac{Z}{(4\pi)^4}\zeta(2)+\mathcal{O}(\epsilon).
\end{aligned}
\tag{5.22}
$$

And for the fermionic mass terms,

$$G_{HHH}^l \supset (X+Y)\left(\frac{\mu}{\pi T}\right)^{4\epsilon} \frac{e^{2\gamma\epsilon}\Gamma\left(\epsilon+\frac{1}{2}\right)^2}{32\pi^5(2\epsilon-1)}(1-2^{-4\epsilon-2})\zeta(4\epsilon+2)$$

$$= -6\frac{X+Y}{(4\pi)^4}\zeta(2)+\mathcal{O}(\epsilon). \tag{5.23}$$

Combining $G_{HHH}^l$ and $D_{HHH}$ we find the fermionic sunset to $\mathcal{O}(T^0)$:

$$\mathbf{I}_F(X,Y,Z)\left(\frac{\mu}{\pi T}\right)^{-4\epsilon}(4\pi)^4 = \tag{5.24}$$

$$(X+Y)\left\{-\frac{1}{2\epsilon^2}-\frac{1+4\gamma}{2\epsilon}+4\gamma_1-2\gamma^2+\frac{3\pi^2}{4}-\frac{3}{2}-2\gamma+4\log^2(2)-6\zeta(2)\right\}$$

$$+Z\left[-\frac{1}{2\epsilon^2}-\frac{1+4\gamma-4\log 4}{2\epsilon}+\right.$$

$$\left.4\gamma_1-2\gamma^2-\frac{3}{2}-\frac{3\pi^2}{4}-2\gamma-8\log^2(2)+16\gamma\log(2)+\log(16)+3\zeta(2)\right].$$

The $\epsilon$ poles agree with previous results [17], and the finite pieces agree with [5].

## 5.3 Comparison of bosonic and fermionic

Let's now compare the complete bosonic and fermionic sunsets at $\mathcal{O}(T^0)$. The bosonic sunset is

$$\mathcal{O}(T^0): \quad \mathbf{I}(X,Y,Z)=4B\times(X+Y+Z)\left[\zeta(2\epsilon+1)^2\right], \tag{5.25}$$

$$B=-\frac{2^{-2(\epsilon+4)}e^{2\gamma\epsilon}\sec(\pi\epsilon)\Gamma(2\epsilon+1)}{\pi^{3/2}\Gamma\left(\frac{3}{2}-\epsilon\right)\Gamma(\epsilon+2)}(2\pi)^{-2}\left(\frac{\mu}{2\pi T}\right)^{4\epsilon}. \tag{5.26}$$

And the fermionic sunset is

$$\mathcal{O}(T^0): \quad \mathbf{I}_F(X,Y,Z)=B_F Z\left[4^{-2\epsilon-1}\left(2^{2\epsilon+1}-1\right)\left(4^{\epsilon+1}-6\right)\zeta(2\epsilon+1)^2\right]$$

$$-B_F(X+Y)\left[2^{-4\epsilon-1}\left(2^{2\epsilon+1}-1\right)^2\zeta(2\epsilon+1)^2\right], \tag{5.27}$$

$$B_F=\frac{\pi^{3/2}4^{1-\epsilon}e^{2\gamma\epsilon}\epsilon\csc(\pi\epsilon)\sec^2(\pi\epsilon)}{\Gamma(2-2\epsilon)\Gamma\left(\frac{1}{2}-\epsilon\right)\Gamma(\epsilon+2)}\frac{1}{(4\pi)^4}\left(\frac{\mu}{\pi T}\right)^{4\epsilon}. \tag{5.28}$$

Note that $B_F=-2^{4\epsilon+3}B$.

Comparing the bosonic and fermionic sunset we confirm that indeed $\mathbf{I}_F(X,X,X)=(4^{1+2\epsilon}-1)\mathbf{I}(X,0,0)$ at $\mathcal{O}(T^0)$: giving a highly non-trivial cross-check. Note that all $\zeta(4\epsilon+2)$ terms cancel between $D_{HHH}$ and $G_{HHH}$, in both the bosonic and the fermionic case (this is required by the IBP identities).

# 6 Numerical Tests

We can now compare the analytical results with numerical caulculation, because the full result is known in terms of definite integrals [17]. The numerical integration is fast when the three masses are of similar order; not when there is a hierarchy between masses.

Start with the bosonic sunset. To ease comparisons we set all masses the same, and then plot $I(X,X,X)/T^2$ versus $\sqrt{X}/T$ in figure 1. Each subplot corresponds to a different scaling $\mu=cT$, as noted in the figures.

In figure 2 we instead focus on the fermionic sunset. In this case we chose the boson mass to be half of the fermion mass, to mark that these are different particles.

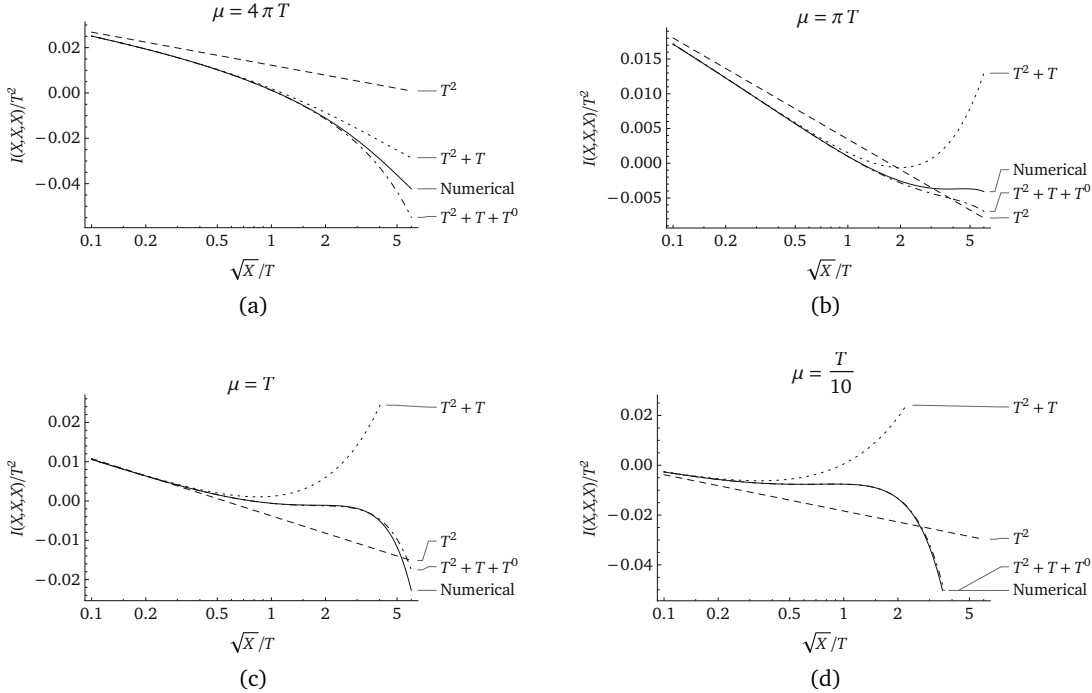

Figure 1: Comparison of the Bosonic high-temperature expansion with the numerical results for various $\mu = cT$ scalings. All masses are equal and the sunset integral is normalized with $T^{-2}$. Comparison when (a) $\mu = 4\pi T$, (b) $\mu = \pi T$, (c) $\mu = T$, and (d) $\mu = T/10$.

## 7 Conclusion

In this paper we have introduced techniques to perform high-temperature expansions in dimensional regularisation. We explicitly derived the $\mathcal{O}(T)$ and $\mathcal{O}(T^0)$ contributions to the bosonic and fermionic thermal sunset integrals, and all contributions of odd powers of $T$ to order $\epsilon^0$. For some previous results, see [18], and see [4,5,12] for IBP calculations of the involved massless master integrals. All terms, at a given order, are expressed in standard functions. These methods, combined with IBP relations, are likely useful even at higher loops. Testing them on 3-loop thermal integrals is an avenue of future research.

Various analytical cross-checks and numerical comparisons validate the results. The previously known approximation to $\mathcal{O}(T^2)$ is, as noted in the past [19], inadequate for any sizeable mass. The next $\mathcal{O}(T)$ correction is important and extends the range of validity considerably. The expansions of the sum-integrals are asymptotic—they depend on several scales [10] and contain terms non-analytic in the masses. However, as can be seen in figures 1 and 2, the expansions are adequate when the first few terms are included.

The bosonic sunset is described well at $\mathcal{O}(T^0)$, for masses up to $\sim 6T$. The range of validity is shorter for the fermionic sunset, where the accuracy is reasonable for masses up to $\sim 3T$. Note that these conclusions depend slightly on the renormalization scale; a clever choice for $\mu$ can extend the range of validity.

There are a number of uses for the result in this paper. In high-temperature calculations where the size of $T$ plays a role in the power counting, such as in EFT- and resummation-techniques, it's not convenient to use the numerical evaluation of the integrals. The method

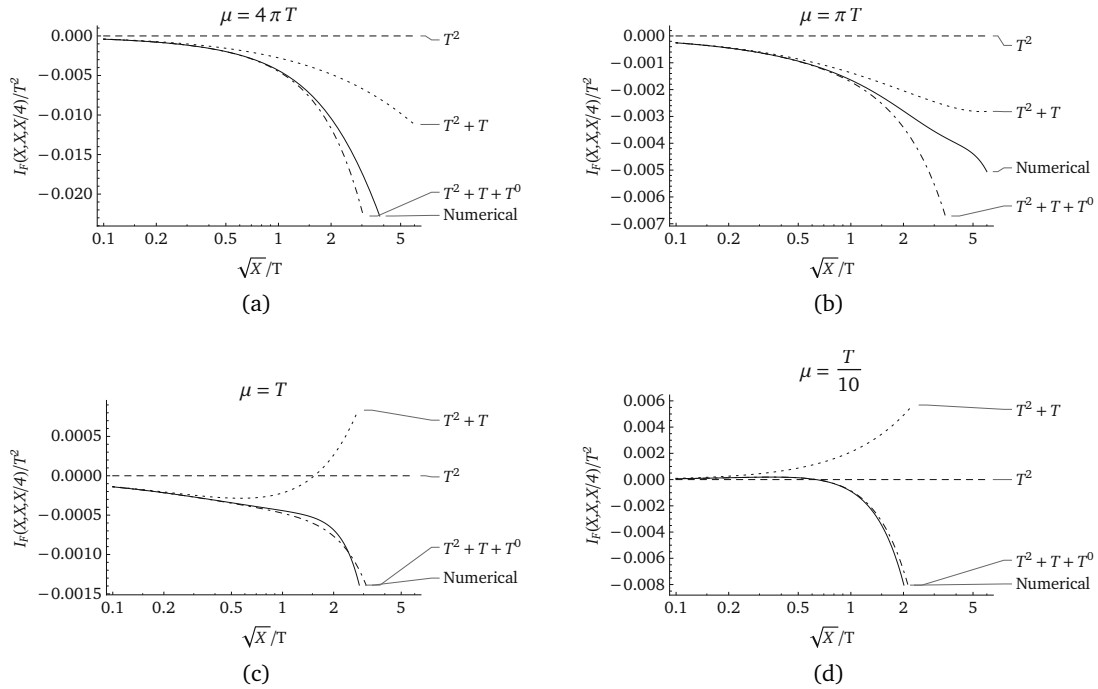

Figure 2: Comparison of the Fermionic high-temperature expansion with the numerical results for various $\mu = cT$ scalings. The two fermionic masses are $\sqrt{X}$, and the bosonic mass is $\frac{\sqrt{X}}{2}$. The sunset integral is normalized with $T^{-2}$. Comparison when (a) $\mu = 4\pi T$, (b) $\mu = \pi T$, (c) $\mu = T$, and (d) $\mu = T/10$.

presented in this paper can extend the reach of perturbation theory.

These results can also be used to improve numerical calculations of the sunset integral itself. As noted in [17], care must be taken when there is a hierarchy between the masses. A numerical calculation can use the expansion we have derived in such a region of parameter space.

# Acknowledgments

We would like to thank Renato Fonseca for many interesting discussions.

**Funding information** The work of A. Ekstedt has been supported by the Grant agency of the Czech Republic, project no. 20-17490S and from the Charles University Research Center UNCE/SCI/013. The research of J. Löfgren was in part funded by the Swedish Research Council, grant no. 621-2011-5107.

# A   Sums & $\zeta$-functions

Many sums used in this paper are of the form

$$\sum_{n,m=1}^{\infty} \frac{1}{n^{2\epsilon+\alpha}(n+m)^{2\epsilon+\alpha}} = \frac{1}{2}\zeta(2\epsilon+\alpha)^2 - \frac{1}{2}\zeta(2\alpha+4\epsilon). \tag{A.1}$$

Sums of this type are straightforward to evaluate [20]. For example, start with the more general sum

$$\sum_{n,m=1}^{\infty}\left(\frac{1}{n^a(n+m)^b}+\frac{1}{n^b(n+m)^a}\right)$$
$$=\sum_{1\le n\le l-1\le\infty}\left(\frac{1}{n^a l^b}+\frac{1}{n^b l^a}\right)=\sum_{n\le l\le\infty}\left(\frac{1}{n^a l^b}+\frac{1}{n^b l^a}\right)-2\zeta(a+b). \tag{A.2}$$

The first sum is symmetric in $n$ and $l$, and so

$$\sum_{n\le l\le\infty}\left(\frac{1}{n^a l^b}+\frac{1}{n^b l^a}\right)=\zeta(a)\zeta(b)+\zeta(a+b). \tag{A.3}$$

Putting everything together,

$$\sum_{n,m=1}^{\infty}\left(\frac{1}{n^a(n+m)^b}+\frac{1}{n^b(n+m)^a}\right)=\zeta(a)\zeta(b)-\zeta(a+b). \tag{A.4}$$

Choosing $a=b$ gives the above sum.

We also need another class of sums for fermionic sunsets. These are of the form

$$\sum_{oo}\frac{1}{n^a(n+m)^b}\equiv\sum_{n\in odd^+, m\in odd^+}\frac{1}{n^a(n+m)^b}. \tag{A.5}$$

And similar for other combinations of even and odd.

These sums are straightforwardly evaluated by using [20, 21]

$$\sum_{oo}\frac{1}{n^a(n+m)^b}=\frac14\sum_{aa}\frac{1}{n^a(n+m)^b}(1-(-1)^n-(-1)^m+(-1)^{m+n}). \tag{A.6}$$

And similarly for related types of sums,

$$\sum_{oe}\frac{1}{n^a(n+m)^b}=\frac14\sum_{aa}\frac{1}{n^a(n+m)^b}(1-(-1)^n+(-1)^m-(-1)^{m+n}), \tag{A.7}$$

$$\sum_{eo}\frac{1}{n^a(n+m)^b}=\frac14\sum_{aa}\frac{1}{n^a(n+m)^b}(1-(-1)^m+(-1)^n-(-1)^{m+n}), \tag{A.8}$$

$$\left(\sum_{eo}+\sum_{oe}\right)\frac{1}{n^a(n+m)^b}=\frac12\sum_{aa}\frac{1}{n^a(n+m)^b}(1-(-1)^{m+n}). \tag{A.9}$$

We also have the relations

$$\left(\sum_{eo}+\sum_{oo}\right)\frac{1}{n^a(n+m)^a}=(1-2^{-a})2^{-a}\zeta(a)^2, \tag{A.10}$$

$$\sum_{oe}\frac{1}{n^a(n+m)^a}=\frac12\left[(1-2^{-a})^2\zeta(a)^2-(1-2^{-2a})\zeta(2a)\right]. \tag{A.11}$$

These more general identities also hold

$$\sum_{eo}\frac{1}{n^a(n+m)^b}+\sum_{oo}\frac{1}{n^b(n+m)^a}=(1-2^{-b})2^{-a}\zeta(a)\zeta(b), \tag{A.12}$$

$$\sum_{oe}\left(\frac{1}{n^a(n+m)^b}+\frac{1}{n^b(n+m)^a}\right)=\left[(1-2^{-a})(1-2^{-b})\zeta(a)\zeta(b)-(1-2^{-a-b})\zeta(a+b)\right]. \tag{A.13}$$

Some finite sums useful at higher-$T$ orders are

$$\sum_{n,m=1}^{\infty} \frac{1}{nm(n+m)^{2\alpha+1}} = (2\alpha+2)\zeta(2\alpha+3) - 2\zeta(2\alpha+1)\zeta(2) - \ldots - 2\zeta(3)\zeta(2\alpha), \quad \text{(A.14)}$$

$$\sum_{n,m=1}^{\infty} \frac{1}{nm(n+m)^{2\alpha}} = \frac{(2\alpha-1)}{2}\zeta(2\alpha+2) - \zeta(2\alpha-1)\zeta(3) - \ldots - \zeta(3)\zeta(2\alpha-1), \quad \text{(A.15)}$$

$$\sum_{n_1,n_2,\ldots n_d=1}^{\infty} \frac{1}{(n_1+n_2+\ldots n_d)^{\alpha}} = \frac{1}{(d-1)!} \sum_z \frac{(z-1)!}{(z-d-2)!} \frac{1}{z^{\alpha}}. \quad \text{(A.16)}$$

## B  Numerical Evaluation of Sums

All sums in this paper of the form $\sum_{n,m=1}^{\infty} \frac{1}{n^a m^b (n+m)^c}$. While all sums needed in this paper can be evaluated in terms of $\zeta$ functions, this is not guaranteed at higher orders. So we here give a prescription to evaluate sums of the given form numerically.

Use the Feynman trick to rewrite the summand,

$$\frac{1}{n^a m^b (n+m)^c} = \int_0^{\infty} dt\, dx\, ds \frac{1}{\Gamma(a)\Gamma(b)\Gamma(c)} t^{a-1} s^{b-a} x^{c-1} \exp\left[-x(n+m) - sm - tn\right]. \quad \text{(B.1)}$$

The sums and the two first integrals give

$$\sum_{n,m=1}^{\infty} \frac{1}{n^a m^b (n+m)^c} = \int dx \frac{x^{c-1} \mathrm{Li}_a(e^{-x}) \mathrm{Li}_b(e^{-x})}{\Gamma(c)}. \quad \text{(B.2)}$$

This integral is intractable in general; yet it turns out that the leading terms, for all sums considered in this paper, come from the $x \to 0$ region; for which the integral is readily evaluated.

Yet higher order $\epsilon$-corrections might be useful in the future. So let's outline how these corrections can be obtained. As an example, consider the sum $\sum_{n,m=1}^{\infty} n^{-1-\epsilon} m^{-1-\epsilon} (n+m)^{-2\epsilon}$. Using the Feynman trick,

$$\sum_{n,m=1}^{\infty} n^{-1-\epsilon} m^{-1-\epsilon} (n+m)^{-2\epsilon} = \int_0^{\infty} dx \frac{x^{2\epsilon-1} \mathrm{Li}_{\epsilon+1}^2(e^{-x})}{\Gamma(2\epsilon)}. \quad \text{(B.3)}$$

Now, there're are two ways to evaluate this integral: the fast way, and the systematic way. The fast way only gives the leading terms in $\epsilon$; the systematic way is more cumbersome but enables one to calculate the sum to an arbitrary order in $\epsilon$.

The fast way uses that $\epsilon$-poles arise from the $x \to 0$ region; so let's introduce a cut-off $R$ and isolate the poles,

$$\int_0^R dx \frac{x^{2\epsilon-1} \mathrm{Li}_{\epsilon+1}^2(e^{-x})}{\Gamma(2\epsilon)} = \frac{1}{6\epsilon^2} + \frac{2\gamma_E}{3\epsilon} + \gamma_E^2 - \frac{\pi^2}{12} - \frac{2\gamma_1}{3} + \mathcal{O}(\epsilon). \quad \text{(B.4)}$$

The systematic way first subtracts the small-$x$ divergences:

$$\frac{x^{2\epsilon-1} \mathrm{Li}_{\epsilon+1}^2(e^{-x})}{\Gamma(2\epsilon)} \underset{x \sim 0}{\approx} \frac{x^{2\epsilon-1} \zeta(\epsilon+1)^2}{\Gamma(2\epsilon)} + \frac{2x^{3\epsilon-1}\zeta(\epsilon+1)\Gamma(-\epsilon)}{\Gamma(2\epsilon)}$$
$$- \frac{2x^{2\epsilon}\zeta(\epsilon)\zeta(\epsilon+1)}{\Gamma(2\epsilon)} - \frac{2x^{3\epsilon}\zeta(\epsilon)\Gamma(-\epsilon)}{\Gamma(2\epsilon)} + \frac{x^{4\epsilon-1}\Gamma(-\epsilon)^2}{\Gamma(2\epsilon)}. \quad \text{(B.5)}$$

The integral is performed with a regulator; we choose the same as in [22]:

$$g_n(x) = (e^{2x})_n e^{-2x} \tag{B.6}$$

$$g_n(x) \underset{x \sim 0}{\approx} 1 + \mathcal{O}(x^{n+1}), \tag{B.7}$$

$$(e^{2x})_n \equiv \sum_{i=0}^{n} \frac{x^i}{i!}, \tag{B.8}$$

where the exponential $e^{-2x}$ is chosen to reproduce the asymptotic behaviour of $\text{Li}^2_{\epsilon+1}(e^{-x})$.

The divergences are then

$$L_{\text{div}}(x) \equiv g_1(x) \frac{x^{2\epsilon-1} \zeta(\epsilon+1)^2}{\Gamma(2\epsilon)} + g_1(x) \frac{2x^{3\epsilon-1} \zeta(\epsilon+1) \Gamma(-\epsilon)}{\Gamma(2\epsilon)}$$
$$- g_0(x) \frac{2x^{2\epsilon} \zeta(\epsilon) \zeta(\epsilon+1)}{\Gamma(2\epsilon)} - g_0(x) \frac{2x^{3\epsilon} \zeta(\epsilon) \Gamma(-\epsilon)}{\Gamma(2\epsilon)} + g_1(x) \frac{x^{4\epsilon-1} \Gamma(-\epsilon)^2}{\Gamma(2\epsilon)}. \tag{B.9}$$

Which gives

$$L_{\text{div}} = \int_0^\infty dx \, L_{\text{div}}(x) = -\frac{2\gamma_1}{3} + \frac{1}{6\epsilon^2} + \frac{2\gamma}{3\epsilon} + \gamma^2 - \frac{\pi^2}{12} + \epsilon \, [-0.650...] + \mathcal{O}(\epsilon^2), \tag{B.10}$$

where the constant number within brackets involves various derivatives of gamma and zeta functions.

The finite part can be numerically integrated and is

$$L_{\text{finite}} = \int_0^\infty dx \left[ \frac{x^{2\epsilon-1} \text{Li}^2_{\epsilon+1}(e^{-x})}{\Gamma(2\epsilon)} - L_{\text{div}} \right] = \epsilon \, [1.676...] + \epsilon^2 \, [-2.656] + \mathcal{O}(\epsilon^3). \tag{B.11}$$

Using this method one can evaluate all possible sums arising in the high-temperature expansion.

## C  Coordinate space propagator

The propagator's Fourier transform is

$$G(R) = \int \frac{d^d p}{(2\pi)^d} \frac{e^{ip \cdot R}}{p^2 + m^2}. \tag{C.1}$$

We'll proceed in two steps. First, the angular integration, and then the "radial" integration. Recall the volume element of a $d$-dimensional sphere

$$\int d^d p = \int_{p=0}^{p=\infty} dp \int_{\phi_1=0}^{\pi} \cdots \int_{\phi_{d-2}=0}^{\pi} \int_{\phi_{d-1}=0}^{2\pi} p^{d-1} \sin^{d-2} \phi_1 \ldots \sin \phi_{d-2} d\phi_1 \ldots d\phi_{d-1}. \tag{C.2}$$

To simplify the calculation, orient the coordinate system so that $\vec{p} \cdot \vec{R} = pR \cos \phi_1$; for the angular integration then splits into one integration over $\phi_1$ times the solid angle for a $d-1$ sphere. Explicitly,

$$\int d^d p = \frac{2\pi^{d/2-1/2}}{\Gamma(d/2-1/2)} \int_{p=0}^{p=\infty} dp \int_{\phi_1=0}^{\pi} p^{d-1} d\phi_1. \tag{C.3}$$

The remaining angular integral is

$$\int_0^\pi \sin(\phi)^{d-2}\, e^{ipR\cos\phi}\,\mathrm{d}\phi = \sqrt{\pi}\left(\frac{2}{pR}\right)^{d/2-1}\Gamma(d/2-1/2)J_{d/2-1}(pR), \qquad (C.4)$$

where $J$ is the Bessel function of the first kind. All that remains is

$$\int_0^\infty \mathrm{d}p\,p^{d-1}\frac{p^{1-d/2}}{p^2+m^2}J_{d/2-1}(pR) = m^{d/2-1}K_{1-d/2}(mR). \qquad (C.5)$$

Finally,

$$G(R) = (2\pi)^{-d/2}\left(\frac{m}{R}\right)^{d/2-1}\left(\frac{e^{\gamma_E}\mu^2}{4\pi}\right)^{\epsilon}K_{1-d/2}(mR), \qquad (C.6)$$

with $K$ being the modified Bessel function of the second kind, and the $\mu$ term added as usual in dimensional regularization.

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
