# Peer review of "The High-Temperature Expansion of the Thermal Sunset"

_SciPost Physics, doi:SciPost Phys. Core 3, 008 (2020)_

## Round 2 · Referee Report · Anonymous (Referee 1) · 2020-7-18

Report
In this paper thermal 2-loop sum-integrals are considered. Going to
the high-temperature limit, the authors derive subleading terms in an
expansion in masses over the temperature.
As far as I can see, the technical computations look sound, and I have
no immediate reason to doubt their correctness.
A much bigger worry is whether the results are novel and represent the
state of the art. Usually, this kind of expressions are reported in an
appendix of a paper whose main focus is on a physics application, so
it is not easy to carry out a comprehensive literature scan. However,
by rapidly checking who has cited the classic hep-ph/9408276,
hep-ph/9410360 by Arnold and Zhai in recent years, I located eq.(A.22)
of 1911.09123, which precedes what the authors claim as a new result
on the last line of their (2.14). Even more importantly, from (A.6) of
1911.09123, I infer that such terms can be given in $d$ dimensions in
closed form, after making use of integration-by-parts (IBP) identities,
so it looks that the authors have missed this modern tool of choice.
Related to the above, the state of the art of massless
high-temperature sum-integrals has been on the 3-loop level since more
than 25 years. Given that the current paper has its sole focus on
technical aspects at the lower 2-loop level, I think that pioneering
references, like Arnold and Zhai, or more recent works by Schroder
[e.g. 1207.5666 and references therein], who introduced IBP
tools for this problem, should be mentioned for context.
On a conceptual note, figs. 1 and 2 suggest that the mass expansion
considered is an 'asymptotic' one (non-convergent in a mathematical
sense). Perhaps the authors could explain why the expansion might
nevertheless be helpful?
Finally, the presentation seems rather careless, with a silly title,
many incomplete sentences (without a verb), overlong lines like the
one above (3.22), colloquial wordings, etc. The authors would be well
advised to try and render their presentation somewhat more 'scientific'.
the high-temperature limit, the authors derive subleading terms in an
expansion in masses over the temperature.
As far as I can see, the technical computations look sound, and I have
no immediate reason to doubt their correctness.
A much bigger worry is whether the results are novel and represent the
state of the art. Usually, this kind of expressions are reported in an
appendix of a paper whose main focus is on a physics application, so
it is not easy to carry out a comprehensive literature scan. However,
by rapidly checking who has cited the classic hep-ph/9408276,
hep-ph/9410360 by Arnold and Zhai in recent years, I located eq.(A.22)
of 1911.09123, which precedes what the authors claim as a new result
on the last line of their (2.14). Even more importantly, from (A.6) of
1911.09123, I infer that such terms can be given in $d$ dimensions in
closed form, after making use of integration-by-parts (IBP) identities,
so it looks that the authors have missed this modern tool of choice.
Related to the above, the state of the art of massless
high-temperature sum-integrals has been on the 3-loop level since more
than 25 years. Given that the current paper has its sole focus on
technical aspects at the lower 2-loop level, I think that pioneering
references, like Arnold and Zhai, or more recent works by Schroder
[e.g. 1207.5666 and references therein], who introduced IBP
tools for this problem, should be mentioned for context.
On a conceptual note, figs. 1 and 2 suggest that the mass expansion
considered is an 'asymptotic' one (non-convergent in a mathematical
sense). Perhaps the authors could explain why the expansion might
nevertheless be helpful?
Finally, the presentation seems rather careless, with a silly title,
many incomplete sentences (without a verb), overlong lines like the
one above (3.22), colloquial wordings, etc. The authors would be well
advised to try and render their presentation somewhat more 'scientific'.

---

## Round 3 · Author Response

We thank the referee for their comments. And the updated version of the paper takes into account their suggestions. As such we have included references to previous work dealing with IBP methods.
Though we were aware of IBP methods in general, we had set them aside because we did not need them. But in retrospect we agree with the referee's assessment that showing a holistic picture is desirable. We now discuss IBP methods, and in this updated version we have made it clear which parts have been calculated before, with appropriate references to the IBP literature.
Regarding the referee's further comments, we agree that the expansion is asymptotic. Some indications of this is that the integral depends on several scales, and that the expansion contains an infinite number of non-analytic terms. The expansion's usefulness stems from that, as with Feynman diagrams, it provides a good approximation for the first couple of terms. We now briefly comment on this in the paper.
Though we have disparate views from the referee as to what constitutes scientific writing, we have taken the referee's suggestions to heart and altered the text accordingly.
We again thank the referee for reading through the manuscript, and for giving insightful comments.
Though we were aware of IBP methods in general, we had set them aside because we did not need them. But in retrospect we agree with the referee's assessment that showing a holistic picture is desirable. We now discuss IBP methods, and in this updated version we have made it clear which parts have been calculated before, with appropriate references to the IBP literature.
Regarding the referee's further comments, we agree that the expansion is asymptotic. Some indications of this is that the integral depends on several scales, and that the expansion contains an infinite number of non-analytic terms. The expansion's usefulness stems from that, as with Feynman diagrams, it provides a good approximation for the first couple of terms. We now briefly comment on this in the paper.
Though we have disparate views from the referee as to what constitutes scientific writing, we have taken the referee's suggestions to heart and altered the text accordingly.
We again thank the referee for reading through the manuscript, and for giving insightful comments.

---

## Round 3 · List of Changes

- Included discussion of IBP methods, with references.
- Altered the text to be more formal.
- Provided brief discussion regarding asymptotic expansion.

---

## Editorial Decision

published